# Machine learning coarse-grained potentials of protein thermodynamics

**Maciej Majewski** [1,2,14], **Adrià Pérez** [1,2,14], **Philipp Thölke** [1], **Stefan Doerr**[2], **Nicholas E. Charron**[3,4,5], **Toni Giorgino** [6], **Brooke E. Husic**[7,8,9,10], **Cecilia Clementi** [3,4,5,11] ✉, **Frank Noé** [5,7,11,12] ✉ & **Gianni De Fabritiis** [1,2,13] ✉

A generalized understanding of protein dynamics is an unsolved scientific problem, the solution of which is critical to the interpretation of the structure-function relationships that govern essential biological processes. Here, we approach this problem by constructing coarse-grained molecular potentials based on artificial neural networks and grounded in statistical mechanics. For training, we build a unique dataset of unbiased all-atom molecular dynamics simulations of approximately 9 ms for twelve different proteins with multiple secondary structure arrangements. The coarse-grained models are capable of accelerating the dynamics by more than three orders of magnitude while preserving the thermodynamics of the systems. Coarse-grained simulations identify relevant structural states in the ensemble with comparable energetics to the all-atom systems. Furthermore, we show that a single coarse-grained potential can integrate all twelve proteins and can capture experimental structural features of mutated proteins. These results indicate that machine learning coarse-grained potentials could provide a feasible approach to simulate and understand protein dynamics.

Proteins are complex dynamical systems that exist in an equilibrium of distinct conformational states, and their multi-state behavior is critical for their biological functions[1–5]. A complete description of the dynamics of a protein requires the determination of (1) its stable and metastable conformational states, (2) the relative probabilities of these states, and (3) the rates of interconversion among them. Here, we focus on addressing the first two problems by demonstrating how to learn coarse-grained potentials that preserve protein thermodynamics.

Due to the structural heterogeneity of proteins and the ranges of time and length scales over which their dynamics occur, there is no

single technique that is able to successfully model protein behavior across the whole spatiotemporal scale. Computationally, the main method to study protein dynamics has traditionally been molecular dynamics (MD). The first MD simulation ever made was carried out in 1977 on the BPTI protein in vacuum, and only accounted for 9.2 picoseconds of simulation time[6]. As remarked by Karplus & McCammon[7], these simulations were pivotal towards the realization that proteins are dynamic systems and that those dynamics play a fundamental role in their biological function[2]. When compared with experimental methods such as X-ray crystallography, MD simulations

[1]Computational Science Laboratory, Universitat Pompeu Fabra, Barcelona Biomedical Research Park (PRBB), Carrer Dr. Aiguader 88, 08003 Barcelona, Spain. [2]Acellera Labs, Doctor Trueta 183, 08005 Barcelona, Spain. [3]Department of Physics, Rice University, Houston, TX 77005, USA. [4]Center for Theoretical Biological Physics, Rice University, Houston, TX 77005, USA. [5]Department of Physics, FU Berlin, Arnimallee 12, 14195 Berlin, Germany. [6]Biophysics Institute, National Research Council (CNR-IBF), 20133 Milan, Italy. [7]Department of Mathematics and Computer Science, FU Berlin, Arnimallee 12, 14195 Berlin, Germany. [8]Lewis Sigler Institute for Integrative Genomics, Princeton University, Princeton, NJ 08540, USA. [9]Princeton Center for Theoretical Science, Princeton University, Princeton, NJ 08540, USA. [10]Center for the Physics of Biological Function, Princeton University, Princeton, NJ 08540, USA. [11]Department of Chemistry, Rice University, Houston, TX 77005, USA. [12]Microsoft Research AI4Science, Karl-Liebknecht Str. 32, 10178 Berlin, Germany. [13]Institució Catalana de Recerca i Estudis Avançats (ICREA), Passeig Lluis Companys 23, 08010 Barcelona, Spain. [14]These authors contributed equally: Maciej Majewski, Adrià Pérez. ✉e-mail: cecilia.clementi@fu-berlin.de; frank.noe@fu-berlin.de; gianni.defabritiis@upf.edu

may obtain a complete description of the dynamics in atomic resolution. This information can explain slow events at the millisecond or microsecond timescale, typically with a femtosecond time resolution.

In the last several decades, there have been many attempts to better understand protein dynamics by long unbiased MD. For example, Lindorff-Larsen et al.[8] and Piana et al.[9] simulated several proteins that undergo multiple folding events over the course of micro- to millisecond trajectories, yielding crucial insights into the hierarchy and timescales of the various structural rearrangements. With current technological limitations, unbiased MD is not capable of describing longer-timescale events, such as the dynamics of large proteins or the formation of multi-protein complexes. Due to the computational cost and timescales involved, there are just a few examples of modeling of such events, including folding of a dimeric protein Top7-CFr[10] and all-atom computational reconstruction of protein-protein (Barnase-Barstar) recognition[11]. Many methods have been developed to alleviate these sampling limitations, for instance, umbrella sampling[12], biased Monte Carlo methods[13], and biased molecular dynamics like replica-exchange[14,15], steered MD[16,17], and metadynamics[18]. More recently, a new generative method based on normalizing flows has been proposed to sample structures from the Boltzmann distribution in one-shot, thereby avoiding the many steps needed in MD to sample different metastable states[19,20].

Another way to access the timescales of slow biological processes is through the use of coarse-graining (CG) approaches. Coarse-graining has a long history in the modeling of protein dynamics[21,22] and since the pioneering work of Levitt and Warshel[23], many different approaches to CG have been proposed[24–29]. Notably, the work by Hills et al.[30] has made significant strides towards creating a transferable bottom-up coarse-grained potential for the simulation of proteins, contributing valuable insights to the field. Popular CG approaches include structure-based models[31], MARTINI[32,33], CABS[34], AWSEM[35], and Rosetta[36]. In general, a CG model consists of two parts: the selection of the CG resolution (or mapping) and the design of an effective energy function for the model once the mapping has been assigned. Although recent work has attempted to combine these two points[37], they are in general kept distinct. The choice of an optimal mapping strategy is still an open research problem[38–40] and we will assume in the following that the mapping is given, focusing instead on the second point, which is the choice of an energy function for the CG model that can reproduce relevant properties of the fine-grained system. Recently, our groups and others have used machine learning methods to extend the theoretical ideas of coarse-graining to systems of practical interest, which provides a systematic and general solution to reduce the degrees of freedom of a molecular system by building a potential of mean force over the coarse-grained system[41–47].

Machine learning models, in particular neural network potentials (NNPs), can learn fast, yet accurate, potential energy functions for use in MD simulations by training on large-scale databases obtained from more expensive approaches[43,44,48–51]. One particularly interesting feature of machine learning potentials is that they can learn many-body atomic interactions[52]. A steady level of improvement of the methodology over the years has led to dozens of novel and better modeling architectures for predicting the energy of small molecules. The first important contributions are rooted in the seminal works by Behler and Parrinello[53] and Rupp et al.[54]. One of the earliest transferable machine learning potentials for biomolecules, ANI-1[55], is based on Behler-Parrinello (BP) representation, while other models use more modern graph convolutions[51,56,57].

In this work, we investigate twelve non-trivial protein systems with a variety of secondary structural elements. We build a unique multi-millisecond dataset of unbiased all-atom MD simulations of studied proteins. We show the recovery of experimental conformations starting from disordered configurations through the classical Langevin simulations of a machine-learned CG force field. We demonstrate transferability across macromolecular systems by using a single multi-protein machine learning potential for all the targets. Finally, we investigate the predictive capabilities of the NNP through simulation and analysis of selected mutants (i.e., sequences outside of the training set).

## Results

### Multi-millisecond all-atom molecular dynamics dataset

We created a large-scale dataset of all-atom MD simulations by selecting twelve fast-folding proteins, studied previously by Kubelka et al.[58] and Lindorff-Larsen et al.[8] Supplementary Table 1. These proteins contain a variety of secondary structural elements, including $\alpha$-helices and $\beta$-strands, as well as unique tertiary structures and various lengths from 10 to 80 amino acids. In the case of the shortest proteins, Chignolin and Trp-Cage (up to 20 amino acids), the secondary structure is quite simple. In general, the dataset contains a higher proportion of $\alpha$-helical proteins. The exceptions are the $\beta$-turn present in Chignolin, the mostly $\beta$-sheet structure of WW-Domain, and the mixed $\alpha\beta$ structures of BBA, NTL9, and Protein G (Fig. 1). The dataset was generated by performing MD on each of the proteins starting from random coil conformations, simulating their whole dynamics and reaching the native structure. The total size of the dataset amounts to approximately 9 ms of simulation time across all proteins (Table 1). The dataset is available for download as a part of Supplementary Information.

### Coarse-grained neural network potentials

A common approach to bottom-up coarse-graining is to seek thermodynamic consistency; i.e., the equilibrium distribution sampled by the CG model—and thus all thermodynamic quantities computable from it, such as folding free energies—should match those of the all-atom model[30]. Popular approaches to train thermodynamically consistent CG models are relative entropy minimization[59] and variational force matching[27,60,61]. The latter has recently been developed into a machine-learning approach to train NNPs to compute the CG energy[43,44].

Let $\mathbb{D}$ be a dataset of $M$ coordinate-force pairs obtained using an all-atom MD force field. Conformations are given by $\mathbf{r}_c \in \mathbb{R}^{3N_c}$, $c = 1, \ldots, M$ and forces by $\mathbf{F}(\mathbf{r}_c) \in \mathbb{R}^{3N_c}$, where $N_c$ is the number of atoms in the system. The number of atoms $N_c$ depends on $c$ as we wish to also have different protein systems in the dataset $\mathbb{D}$. We define a linear mapping $\Xi$ which reduces the dimensionality of the atomistic system $\mathbf{x} = \Xi\mathbf{r} \in \mathbb{R}^{3n}$, where $3n$ are the remaining degrees of freedom. For example, $\Xi$ could be a simple map to $\alpha$-carbon atom coordinates for each amino acid, to backbone coordinates or to the center of mass. We seek to obtain $U(\mathbf{x}_c; \boldsymbol{\theta}) : \mathbb{R}^{3n} \to \mathbb{R}$ for any configuration $c$ parameterized in $\theta$, such that to minimize the loss

$$L(\mathbf{R}; \boldsymbol{\theta}) = \frac{1}{3nM} \sum_{c=1}^{M} \| \Xi\mathbf{F}(\mathbf{r}_c) + \nabla U(\Xi\mathbf{r}_c; \boldsymbol{\theta})\|^2 \qquad (1)$$

In order to reduce the conformational space accessible during the CG simulation and prevent the system from poor exploration, it is important to provide a prior potential[44,62]. This also serves to reduce the complexity of the force field learning problem, and can equivalently be viewed as imposing physical biases from domain knowledge. The NNP is therefore performing a delta-learning between the all-atom forces and the prior forces. We applied bonded and repulsive terms to avoid rupture of the protein chain as well as clashing beads (Eqs. (11) and (12) in "Methods"). Furthermore, we enforce chirality by introducing a dihedral prior term (Eq. (13) in "Methods"). This prevents the CG proteins from exploring mirror images of the native structures. The functional forms and parameters of all prior terms are available in "Methods".

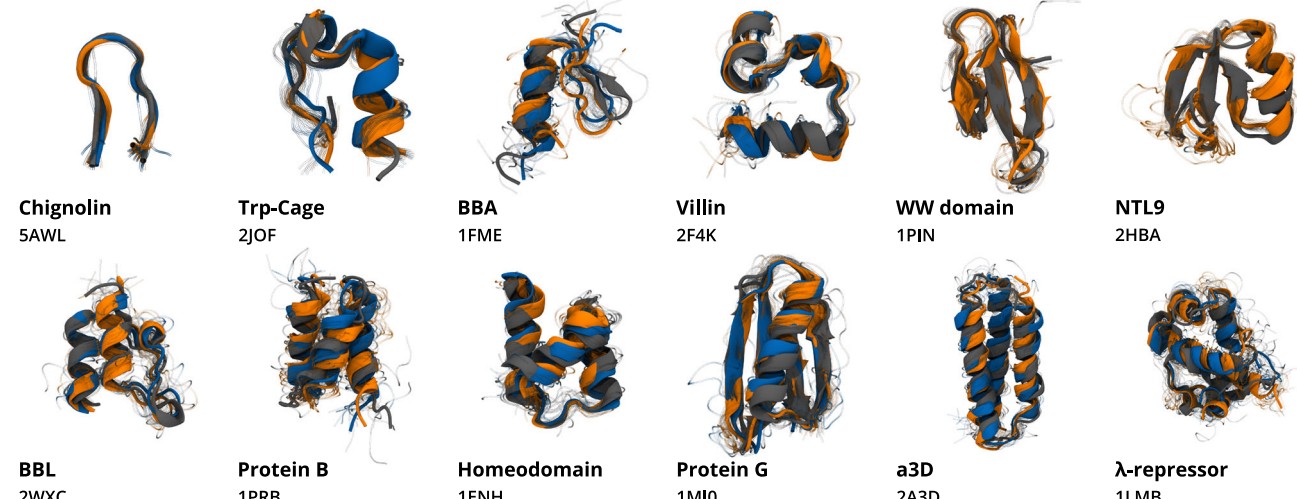

**Fig. 1 | Comparison of simulated and experimental protein structures.** Structures obtained from CG simulations of the protein-specific model (orange) and the multi-protein model (blue), compared to their respective experimental structures (gray). Structures were sampled from the native macrostate, which was identified as the macrostate containing the conformation with the minimum RMSD with respect to the experimental crystal structure. Ten conformations were sampled from each conformational state (visualized as transparent shadows) and the lowest RMSD conformation of each macrostate is displayed in cartoon representation, reconstructing the backbone structure from $\alpha$-carbon atoms. The native conformation of each protein, extracted from their corresponding crystal structure is shown in opaque gray. The text indicates the protein name and PDB ID for the experimental structure. WW-Domain and NTL9 results for the multi-protein model are not shown, as the model failed to recover the experimental structures. The statistics of native macrostates are included in Table 2.

CG representations were created by retaining only certain atoms of each protein's all-atom representation; the retained atoms are referred to as CG beads. NNPs were trained to predict forces based on the coordinates and identities of the beads, where the latter is represented as an embedding vector. Each CG bead comprises the $\alpha$-carbon atom of its amino acid, and each amino acid was described by a unique bead type. In previous work, we experimented with both $\alpha$-carbon and $\alpha\beta$-carbon representation; however, the simpler $\alpha$-carbon representation was sufficient to learn the dynamics of small proteins[63].

## Coarse-grained molecular dynamics with neural network potentials reconstructs the dynamics of proteins

Initially, we carried out CG simulations of all twelve proteins using the models trained on individual all-atom MD datasets corresponding to each protein; that is, we trained twelve models, each one only using the corresponding data for one protein. To validate the models, we performed 32 parallel coarse-grained simulations for each target, starting from conformations sampled across the reference free energy surface, built based on all-atom MD (Supplementary Fig. 1). The intent was to explore the conformational dynamics, sample the native structure and reconstruct the reference free energy surface.

A Markov state model (MSM)[64–68] analysis of CG simulations shows that all of the individual protein models were able to recover the experimental structure of the corresponding target (Fig. 1), accurately predicting all the secondary structure elements and the tertiary structure, with loops and unstructured terminal regions being the most variable parts. For the simplest target, Chignolin, the average root-mean-square deviation (RMSD) value of the native macrostate was 0.7 Å. For less trivial structures, such as WW-Domain or NTL9, the values were below 2.5 Å. For even more complex arrangements of secondary elements, like Protein B and $\lambda$-repressor, the average RMSD of the native macrostate predicted by the network increased to 5.5 and 4.2 Å, respectively. In all cases, however, the network was able to sample conformations below 2.5 Å and global distance test (GDT)[69] scores above 60 (Table 2 and Supplementary Table 2).

For all protein-specific models, simulations were able to sample folding events, in which the protein goes from a random coil to a native conformation (Fig. 2 and Supplementary Fig. 2). The dynamics of transitions is accelerated more than three orders of magnitude, as the process happens in nanosecond timescale, in contrast to microseconds in the case of all-atom MD[8]. It is worth noting that, with current software, coarse-grained molecular dynamics with neural network potentials is 1–2 orders of magnitude slower than equivalent simulation with explicit solvent using classical force fields[63]. However, we expect that this difference is going to reduce fast. In addition, individual trajectories were able to explore the conformational landscape and transition between different metastable states observed in the original all-atom trajectories. For each protein, a representative trajectory is shown in a video included in Supplementary Information (Supplementary Table 3). A few models, in particular Homeodomain, $\alpha$3D, and $\lambda$-repressor, failed to sample direct transitions from ordered to disordered conformations (Supplementary Fig. 2). This could have been caused partially by the model over-stabilizing the native structure.

**Table 1 | All-atom MD simulation dataset generated for this work and used for training and testing of NNPs**

| Protein | Sequence length (#aa) | Aggregated time (μs) | Min. RMSD (Å) |
|---|---|---|---|
| Chignolin | 10 | 186 | 0.15 |
| Trp-Cage | 20 | 195 | 0.45 |
| BBA | 28 | 362 | 1.13 |
| WW-Domain | 34 | 1362 | 0.73 |
| Villin | 35 | 234 | 0.47 |
| NTL9 | 39 | 776 | 0.32 |
| BBL | 47 | 677 | 1.55 |
| Protein B | 47 | 608 | 1.19 |
| Homeodomain | 54 | 198 | 0.56 |
| Protein G | 56 | 2266 | 0.55 |
| $\alpha$3D | 73 | 768 | 1.81 |
| $\lambda$-repressor | 80 | 1422 | 0.82 |

**Table 2 | Native macrostate statistics from all MSMs built with CG simulations from all protein-specific models and the multi-protein model**

| Protein | Protein-specific | | | Multi-protein | | | Reference | | |
|---|---|---|---|---|---|---|---|---|---|
| | Macro prob. (%) | Mean RMSD (Å) | Min RMSD (Å) | Macro prob. (%) | Mean RMSD (Å) | Min RMSD (Å) | Macro prob. (%) | Mean RMSD (Å) | Min RMSD (Å) |
| Chignolin | 19.7 ± 0.8 | 0.7 ± 0.4 | 0.2 | 33.4 ± 0.6 | 1.2 ± 0.6 | 0.2 | 57.5 ± 0.6 | 1.0 ± 0.4 | 0.1 |
| Trp-Cage | 93.2 ± 0.7 | 2.8 ± 0.5 | 1.0 | 81.1 ± 12.0 | 2.9 ± 0.5 | 1.0 | 30.1 ± 3.9 | 2.5 ± 0.8 | 0.4 |
| BBA | 41.1 ± 1.8 | 3.8 ± 1.0 | 1.6 | 17.5 ± 1.4 | 4.4 ± 1.0 | 1.6 | 5.24 ± 0.9 | 3.9 ± 1.3 | 1.1 |
| WW-Domain | 15.4 ± 2.5 | 2.5 ± 0.5 | 1.1 | — | — | — | 45.5 ± 1.1 | 2.7 ± 1.1 | 0.7 |
| Villin | 77.3 ± 8.9 | 2.7 ± 0.9 | 0.8 | 77.7 ± 13.0 | 2.9 ± 0.9 | 1.0 | 69.2 ± 1.4 | 3.4 ± 1.8 | 0.5 |
| NTL9 | 32.0 ± 2.2 | 2.4 ± 0.9 | 0.6 | — | — | — | 15.3 ± 3.5 | 1.6 ± 0.9 | 0.3 |
| BBL | 95.0 ± 0.5 | 2.8 ± 1.2 | 1.0 | 47.8 ± 8.3 | 2.4 ± 0.6 | 0.9 | 30.5 ± 2.7 | 3.1 ± 1.3 | 0.7 |
| Protein B | 71.6 ± 1.6 | 5.6 ± 1.0 | 2.3 | 75.8 ± 6.4 | 3.3 ± 0.5 | 2.0 | 30.1 ± 0.4 | 4.4 ± 1.4 | 1.2 |
| Homeodomain | 77.6 ± 14.0 | 2.8 ± 0.4 | 1.8 | 98.5 ± 0.4 | 2.4 ± 0.3 | 1.5 | 53.5 ± 1.9 | 2.3 ± 1.5 | 0.3 |
| Protein G | 64.8 ± 3.9 | 2.7 ± 0.5 | 1.4 | 2.1 ± 0.9 | 2.2 ± 0.4 | 1.2 | 17.1 ± 1.6 | 2.9 ± 1.9 | 0.6 |
| α3D | 90.5 ± 6.9 | 3.2 ± 0.2 | 2.4 | 96.4 ± 2.4 | 3.4 ± 0.3 | 2.2 | 67.9 ± 1.2 | 3.5 ± 0.7 | 1.8 |
| λ-repressor | 77.4 ± 10.7 | 4.3 ± 0.5 | 2.1 | 79.1 ± 7.0 | 4.6 ± 0.7 | 2.8 | 21.9 ± 0.5 | 4.5 ± 1.2 | 0.8 |

The data describes the identified native macrostate for each protein, showing equilibrium probabilities in percentage (Macro prob.), average (with standard deviation), and minimum RMSD values with respect to the experimental structure.

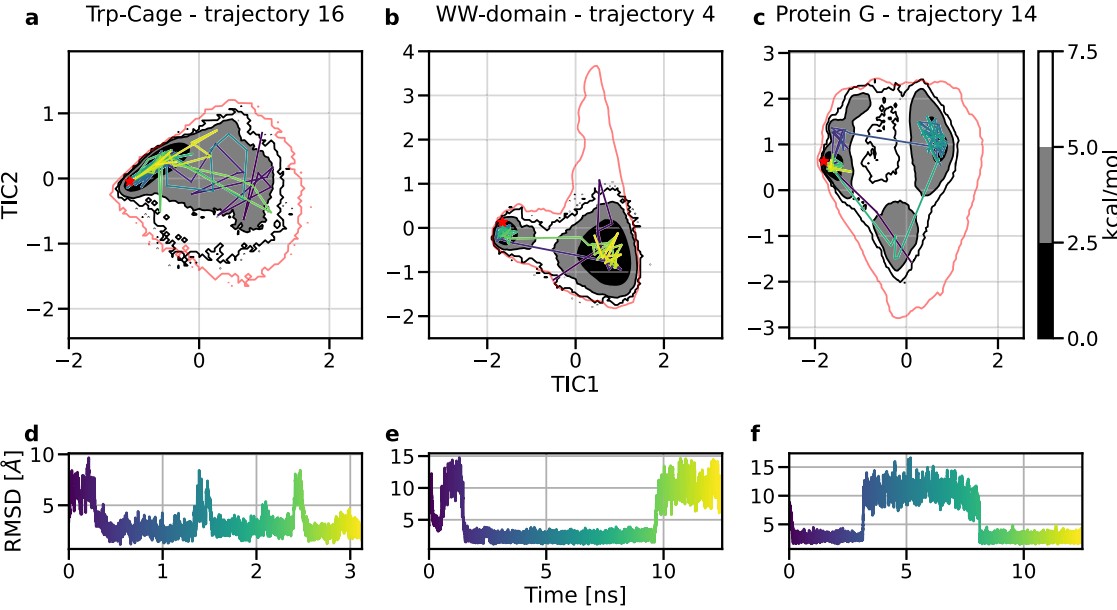

**Fig. 2 | Trajectory analysis of protein dynamics.** Three individual CG trajectories selected from validation MD of Trp-Cage, WW-Domain, and Protein G. Each visualized simulation, colored from purple to yellow, explores the free energy surface, accesses multiple major basins and transitions among conformations. Top panels: 100 states sampled uniformly from the trajectory plotted over CG free energy surface, projected over the first two time-lagged independent components (TICs) for Trp-Cage (**a**), WW-Domain (**b**), and Protein G (**c**). The red line indicates the all-atom equilibrium density by showing the energy level above the free energy minimum with the value of 7.5 kcal/mol. The experimental structure is marked as a red star. Bottom panels: Cα-RMSD of the trajectory with reference to the experimental structure for Trp-Cage (**d**), WW-Domain (**e**), and Protein G (**f**). Source data are provided as a Source data file.

## Coarse-grained potentials maintain the energetic landscape

In order to estimate the equilibrium distribution and approximate the free energy surfaces from the CG simulations, we built MSMs for each CG simulation set. Time-lagged independent component analysis (TICA)[70,71] was used to project coarse-grained trajectories onto the first three components, using covariances computed from reference all-atom MD. Overall, the MSMs were able to recover the surface describing the dynamics, correctly locating the position of the global minimum in the free energy surface for all cases except Protein B (Fig. 3 and Supplementary Figs. 3 and 4). The most ill-defined regions of TIC space correspond to unstructured conformations, which are more difficult for the models to sample. In most of the models, simulations transition rapidly to the native structure, and only the surface around the global minimum is sampled. This is particularly true for larger helical proteins, such as Homeodomain, α3D, and λ-repressor, where the space explored falls mostly around the native structure. Alternatively, in Chignolin, Trp-Cage, Villin, NTL9, and Protein G, the models are able to sample most of the free energy surface, locating all different metastable minima identified through TICA.

In the case of Protein G, the model was able to identify all the metastable states, sharing similar features as the reference all-atom MD simulations (Fig. 4). Furthermore, the model correctly replicates the main transition to the native structure and allows for a possible interpretation of the folding pathway. In the most probable folding pathway, the protein initially forms an intermediate, partially folded state containing the α-helix and the first hairpin. Next, the native

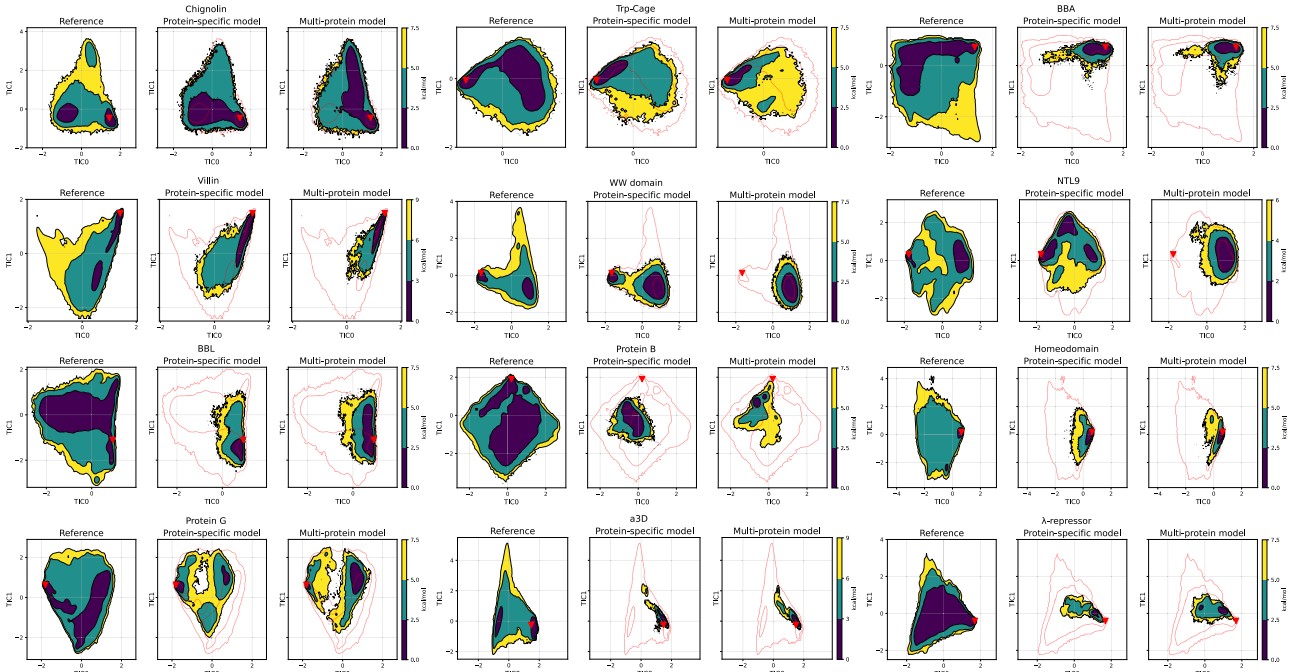

**Fig. 3 | Free energy surface comparison across all-atom reference and coarse-grained models.** Comparison between the reference MD (left), protein-specific model (center), and multi-protein model (right) coarse-grained simulations free energy surface across the first two TICA dimensions for each protein. The free energy surface for each simulation set was obtained by binning over the first two TICA dimensions, dividing them into a 80 × 80 grid, and averaging the weights of the equilibrium probability in each bin computed by the Markov state model. The red triangles indicate the experimental structures. The red line indicates the all-atom equilibrium density by showing the energy level above free energy minimum with the values of 9 kcal/mol for Villin and $\alpha$3D, 6 kcal/mol for NTL9, and 7.5 kcal/mol for the remaining proteins. Source data are provided as a Source data file.

structure is completed by the formation of the second hairpin. Alternatively, a second pathway is possible where the structure goes through a misfolded state with an almost complete native structure except for the first hairpin, which shows increased flexibility. This replicates the results of all-atomistic MD simulations performed by Lindorff-Larsen et al.[8]. The variant simulated both there and in this study is intermediate in sequence between the wild type and redesigned NuG2 variant. Despite high similarities in the sequence, experiments show that these variants exhibit distinct folding pathways. The difference is in the order of formation of the elements of $\beta$-sheet; in the wild-type variant of Protein G, the second hairpin folds before the first hairpin[72,73] while in the NuG2 variant the order is reversed[74]. The CG simulation using NNP shows the majority of flow going into the NuG2 variant folding, which agrees with one of the possible folding pathways. In addition, the simulation correctly recovered the minima around the native conformation of Protein G, however, the position of the other minima on the free energy surface are less similar. In general, the force-matching method does not preserve kinetics[27,60], so the height of the energy barriers is not expected to be accurately captured, as shown in the free energy plot (Supplementary Figs. 3 and 4).

For NTL9, the model correctly replicates the transition to the native structure, allowing for a possible interpretation of its pathway (Supplementary Fig. 5). From the structural samples, we can see that the $\alpha$-helix is the first secondary element formed that appears even in the unstructured macrostate. By identifying the intermediate state, where the $\beta$-sheet is not entirely formed, we can understand that $\beta$-sheet formation is the limiting step in the process.

### The multi-protein model recovers the native structures of most reference proteins

The individual CG models recovered native structures of the proteins, demonstrating the success of our approach for complex structures.

These NNPs are, however, limited to the individual targets they were trained on. In the next step, we examined if it was possible to train a single, multi-protein model using the reference simulation data of all the protein targets (Supplementary Fig. 6). We then simulated all targets with the multi-protein model, in the same way we did for the protein-specific models. The main objective of the multi-protein model is to match the results of individual models using a single CG potential.

The CG simulations show that the multi-protein model is able to reproduce the native structure of most of the proteins, with the exception of NTL9 and WW-Domain (Fig. 1 and Supplementary Figs. 3 and 4). We identify each native macrostate based on its RMSD to the corresponding experimental structure. However, a simple criterion of minimal potential energy produced by the NNP is able to correctly identify all of the native macrostates for protein-specific models described in the previous section, and in nine out of ten cases (excluding NTL9 and WW-Domain) where the multi-protein model sampled the native structure. The only exception is BBA, where a quasi-folded macrostate is selected instead which has not fully stabilized the small $\beta$-sheet (Supplementary Fig. 7).

In general, the free energy landscapes produced by the multi-protein model resemble the protein-specific ones. However, the multi-protein model neglects energetic barriers and overestimates the global minima, which leads to some trajectories being stuck at the native structure (Fig. 3 and Supplementary Figs. 3, 4 and 8).

In the cases of NTL9 and WW-Domain, the native structure is sampled only as an artefact of starting positions being equally distributed on the reference free energy surface (Supplementary Fig. 1). The native structure is not stable as all simulations move quickly to unstructured conformations. For Protein G, simulations show that the native conformation is stable, but we could not sample any transitions into this conformation from random coil initial conditions, although we could capture unfolding events (Supplementary Fig. 9). In these

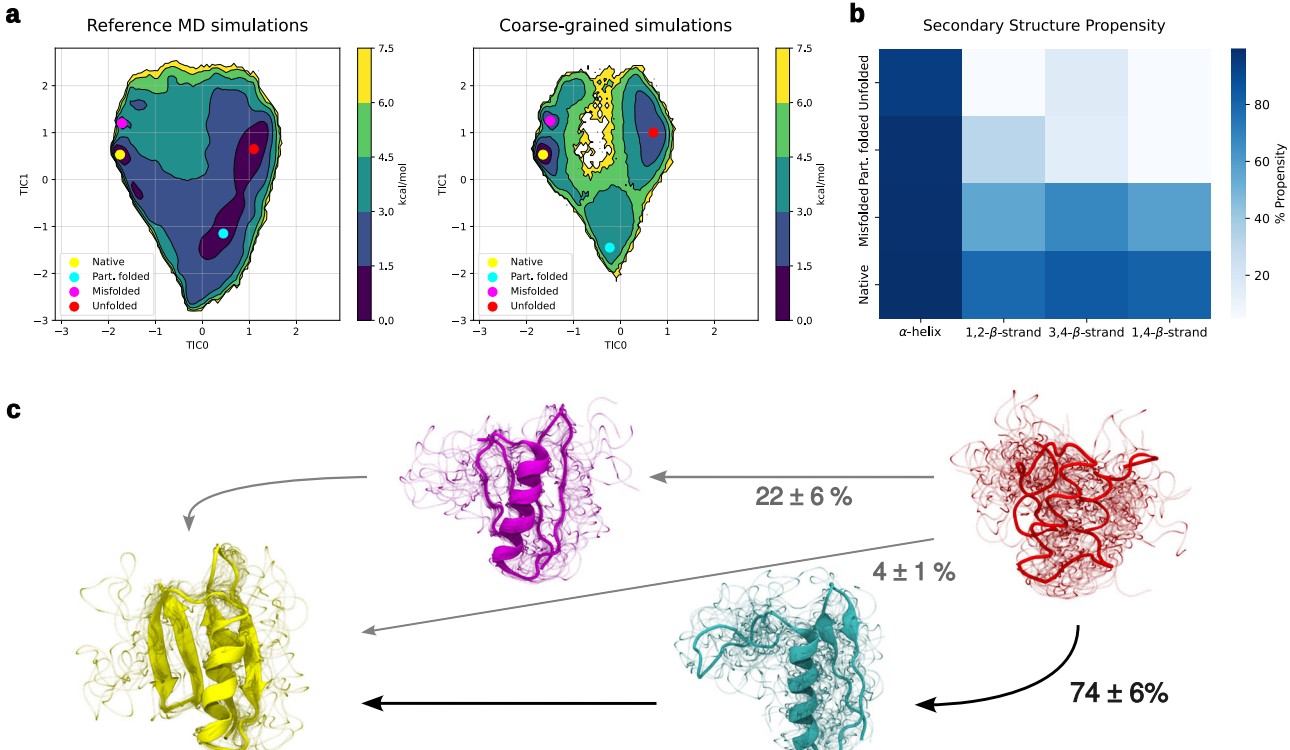

**Fig. 4 | Free energy surface and structural analysis of Protein G simulations.**
**a** Free energy surface of Protein G over the first two TICs for the all-atom MD simulations (top) and the coarse-grained simulations (bottom) using the protein-specific model. The circles identify different relevant minima (yellow−native, magenta−misfolded, cyan−partially folded, red−random coil). **b** The propensity of all the secondary structural elements of Protein G across the different macrostates, estimated using an RMSD threshold of 2 Å for each structural element shown in the x-axis. **c** Sampled conformations from the macrostates of coarse-grained simulations corresponding to the marked minima in the free energy surfaces in (**a**). Sampled structure colors correspond to the minima colors in the free energy surface plot, with blurry lines of the same color showing additional conformations from the same state. Arrows represent the main pathways leading from the random coil to the native structure with the corresponding percentages of the total flux of each pathway. Source data are provided as a Source data file.

cases, the native structure is identified as the lowest energy structure by the NNP. Therefore we can promote transitions to the low-energy states by lowering the temperature of the simulation. We simulated these systems at temperatures of 300 K and 250 K. This approach showed that the NNP recovers the native structure of NTL9 at 300 K. For Protein G and WW-Domain, lowering the temperature stabilizes conformations that resemble the experimental structures, but we have not observed transitions from fully disordered to ordered structures (Supplementary Fig. 10).

One aspect that the failed cases have in common is the presence of β-sheets, which could be the reason why the multi-protein models make the proteins' structured states unstable. Ten out of twelve proteins in the training set contain α-helices, with only Chignolin and WW-domain representing completely β-sheet proteins and BBA, NTL9 and Protein G containing a mix of secondary structure elements. Therefore, the multi-protein model might be biased towards helical structures. Another explanation could be that due to the locality of interactions α-helices may be easier to learn for the NNP. α-helices can be formed gradually with smaller energy barriers, while β-sheets arrange a full strand at a time. In addition, α-helices are stabilized by residues close in the sequence, which provides molecular context even in the random coil state. On the contrary, the stabilizing interactions of β-sheets occur between the residues distant in the sequence. Therefore, for random conformations, the beads are usually outside of the 12 Å upper cutoff of the NNP, reducing the number of examples to learn from. Extending the upper cutoff leads, however, to noisy potentials and an overall worse performance. Similar difficulties with β-sheet proteins were observed during hyperparameter optimization of single-protein NNPs.

For all the helical proteins, the multi-protein model performs similarly to the protein-specific models (Table 2). In some cases, the frequency of transitions between states is altered, as well as the stability of the macrostates, but both models successfully recover the native conformations.

In the case of Trp-Cage, the multi-protein potential outperforms the protein-specific model. The location and the shape of the global minimum match better the reference simulations as well as experimental data, which indicates that the model benefits from additional data from other proteins (Fig. 3 and Supplementary Figs. 3, 4 and 8). In the case of Protein B, the multi-protein model also outperforms the protein-specific one, as it is able to improve the average RMSD of the native macrostate and samples the correct location of the experimental structure, although it is not detected as a minimum (Table 2, Fig. 3).

The results obtained with the multi-protein model are in line with protein-specific models, which indicates that our approach could scale to create a general-use CG force field. This model was able to simulate the transition from random coil to the correct native conformation for almost all target proteins, with the exception of β-sheet proteins (WW-Domain, NTL9, and Protein G), which required simulations at lower temperatures to recover the native state.

**The multi-protein NNP recovers the native structure of mutated proteins**
To further test the multi-protein NNP and assess its predictive power we simulated mutants of the originally targeted proteins. All mutants were sourced from PDB and the mutations did not affect the native structure of the target. Supplementary Table 4 summarizes the

**Table 3 | Native macrostate statistics of mutant variants of the proteins based on the CG simulations performed with the multi-protein model**

| Protein | PDB | Number of substitutions | Min RMSD (Å) | Mean RMSD (Å) | Eq. prob. (%) |
|---------|-----|------------------------|--------------|---------------|---------------|
| BBL | 1BAL | 3 | 1.5 | 3.9 ± 0.9 | 52.3 ± 1.4 |
| Protein B | 1GAB | 2 | 2.3 | 4.9 ± 1.2 | 32.5 ± 1.4 |
| Protein B | 2N35 | 10 | 4.0 | 9.3 ± 1.4 | 19.4 ± 0.8 |
| Homeodomain | 1DU0 | 1 | 1.6 | 2.6 ± 0.4 | 57.0 ± 3.2 |
| Homeodomain | 1P7I | 1 | 1.4 | 2.5 ± 0.3 | 92.2 ± 13.3 |
| Homeodomain | 1P7J | 1 | 3.8 | 4.6 ± 0.3 | 16.6 ± 6.6 |
| Homeodomain | 2HOS | 4 | 1.6 | 3.1 ± 0.8 | 65.4 ± 5.7 |
| Homeodomain | 6M3D | 2 | 1.5 | 2.5 ± 0.3 | 24.5 ± 4.0 |
| $\alpha$3D | 2MTQ | 3 | 2.8 | 4.4 ± 0.6 | 96.2 ± 0.9 |
| $\lambda$-repressor | 1LLI | 3 | 3.1 | 5.1 ± 0.9 | 97.0 ± 0.8 |
| $\lambda$-repressor | 3KZ3 | 6 | 2.3 | 5.8 ± 1.4 | 76.1 ± 7.6 |

The table shows the protein name, PDB ID and the number of amino acid substitutions for each mutant. Results show the minimum RMSD with respect to the mutant experimental structure, as well as the mean RMSD and equilibrium probability of the native macrostate, obtained from an MSM built based on CG simulations of the mutant.

structures selected for the experiment. For each mutant, we initially performed a CG simulation of a single trajectory that started from the native structure using the multi-protein model. In the majority of cases, the structure immediately transitioned to a random coil. The mutants that kept the native conformation for 1 ns were further evaluated using the same protocol we used for the previous CG simulations.

The results show that the multi-protein CG model is able to recover the native conformation for all cases that succeeded in the initial validation, except one (Protein B mutant 2N35), with reasonably low RMSD values (Table 3, Supplementary Fig. 11). Although the NNP was able to simulate the protein dynamics, the exploration of conformational space was limited, as the simulations converge rapidly to the native structure or the conformations resembling it (Supplementary Fig. 12). These cases demonstrate some ability of the multi-protein model to generalize outside of the training set even with a narrow training set of only twelve proteins.

All the examples that recovered the native conformation had very few mutations and were solely helical structures, for which the multi-protein model performs well. In the case of a mutant of Protein B (PDB: 2N35), the NNP failed to obtain the native structure. Its sequence contains 10 mutated residues, which may exceed the capacity of the model to generalize outside of the training set. As shown in Supplementary Table 4, an increased number of mutations reduces the stability of the native macrostate. In the case of $\beta$-sheet containing proteins, even with a point mutation, the model failed to recover the native structure and the amino-acid chains immediately formed unstructured bundles. This observation is not surprising, given the difficulties encountered by the multi-protein model on the $\beta$-sheet containing targets.

Overall, the mutagenesis tests have shown limited but encouraging results for the predictive capabilities of the multi-protein model. Despite its failure to keep the native conformation stable for the sequences that are substantially altered or for proteins that contain $\beta$-sheets, the NNP recovered native macrostates of $\alpha$-helical proteins with minor changes in the sequence. This shows some capacity of the model to generalize.

## Discussion

In the previous work, we have shown that an NNP with a non-transferable coarse-grained model architecture can learn the thermodynamics of a single protein[44]. Then, in the following publication, we replicated the task using a model architecture that is in principle transferable[43]. In this work, we apply a revised model architecture and show that we can effectively learn thermodynamics for twelve structurally diverse proteins at once, in a single model. This demonstrates for the first time that the model architecture is truly transferable and might generalize providing enough data. To achieve that, we generated a multi-millisecond dataset of MD simulations sampling the dynamical landscape of the proteins and used it to obtain machine-learned CG potentials for studying the protein dynamics. Results show that we were able to model protein dynamics in computationally accessible timescales, and recover the native structure of all twelve proteins through coarse-grained MD simulations using NNPs and an $\alpha$-carbon CG representation, with a unique bead type corresponding to each amino-acid type. From the model-generated CG simulation data, we were able to reconstruct multiple metastable states, capturing the folding pathways and the formation of different types of secondary and tertiary structures. In contrast to novel deep-learning structure prediction methods[75,76], our method offers a substantial improvement and models protein dynamics, which is essential for understanding protein function.

The multi-protein model, trained over all proteins in the MD dataset, demonstrates that we were able to model multiple proteins with a single NNP. The following tests on mutants of the 12 proteins have shown the robustness of the multi-protein NNP to small differences in sequence. We highlight, however, that the current training set, while being one of the largest ever produced, only contains data for 12 small proteins. With such a small number of training examples, it is unrealistic to expect that the NNP will model sequences different from the training set. Therefore we do not provide a hold-out test set. While it is not a physical model, this work is a fundamental step in that direction.

There are a few limitations to the current approach. In general, machine learning potentials do not extrapolate well outside of the training set for atom positions that are never sampled in the training set. Therefore, unseen positions are assigned unrealistically low energies and often produce spikes in forces. This has been solved by limiting the physical sampled space with the use of basic prior energy terms[44]. The network also relies on large datasets of all-atom molecular dynamics trajectories which are expensive to produce. Furthermore, the current accuracy of coarse-grained MD is limited by the accuracy of the underlying all-atom simulations. While all-atom force fields are reasonably good for proteins, improved approaches are required for coarse-grained small molecules[77]. Ultimately, the ability to create a truly general model that is transferable from smaller to larger proteins would revolutionize the field[78]. Some works suggest that transferability can be achieved by a sufficient sampling of various configurations and state variables[79]. We think that, in order to learn transferable potentials, some key improvements need to be made, mainly: much larger molecular simulation datasets, alternative training strategies, improved coarse-grained mapping strategies, and more robust architectures that can deal with non-physical states. Current results indicate that this might be achievable.

## Methods

### All-atom molecular dynamics simulations and training data

All initial structures were solvated in a cubic box and ionized as described by Lindorff-Larsen et al.[8]. MD simulations were performed with ACEMD[80] on the GPUGRID.net distributed computing network[81]. The systems were simulated using the CHARMM22*[82] force field and TIP3P water model[83] at the temperature of 350 K. All the simulations were performed following a previously used adaptive sampling strategy[84], in order to explore efficiently as many conformations as possible. Homeodomain dataset also contains simulations that started from the native conformation, as low RMSD values (≤2 Å) with respect

to the native structure are difficult to sample when starting from random coil conformations. A Langevin integrator was used with a damping constant of 0.1 ps⁻¹. Integration time step was set to 4 fs, with heavy hydrogen atoms (scaled up to four times the hydrogen mass) and holonomic constraints on all hydrogen-heavy atom bond terms[85]. Electrostatics were computed using Particle Mesh Ewald with a cutoff distance of 9 Å and grid spacing of 1 Å. Ten NVT simulations of 1 to 10 ns length were carried out for each protein, with a dielectric constant of 80 and temperature of 500 K to generate ten different starting random coil conformations for the production runs. Production simulations consisted of thousands of short trajectories of 20, 50, or 100 ns, distributed across different epochs using the adaptive sampling[84,86] protocols implemented in HTMD[87]. In adaptive sampling, multiple rounds of simulations are performed, and in each round the available trajectories are analyzed to select the initial coordinates for the next round of simulations. The MSM constructed during the analysis was done using atom distances, using TICA for dimensionality reduction and k-centers for clustering. From the trajectories, we extracted forces and coordinates with an interval of 100 ps. Total aggregate times used for training for all the proteins are summarized in Table 1.

Based on the MD dataset we built MSMs for each protein. The models were able to describe the conformational dynamics of each protein, sample the native conformation and identify intermediate and metastable states for some of them, such as Villin, NTL9, WW-Domain, or Protein G (Fig. 3).

## Neural network training

To train NNPs we used TorchMD-Net[77]. We performed an exhaustive hyperparameter search, which is described in Supplementary Table 5. The total number of parameters of the network is 294,565. The data was randomly split between training (85%), validation (5%), and testing (10%). An epoch for simulation was selected when the validation loss reached a minimum or a plateau. The training and validation loss reported as MSE loss, test loss reported as L1 loss and the learning rates of models selected for simulation are presented in Supplementary Figs. 6 and 13. The models were trained using Nvidia GeForce RTX 2080 graphics cards. The training of protein-specific models took from 7 min/epoch on a single GPU for Chignolin to 24 min/epoch on 2 GPUs for $\lambda$-repressor. The training of the multi-protein model took 46 min/epoch on 3 GPUs.

A lot of effort was dedicated to building a graph neural network architecture TorchMD-GN, inspired by SchNet[56,88] and PhysNet[51] and optimized to work optimally on noisy forces and energies proper of the reduced dimensionality of our coarse-graining. This scenario is different from the quantum case, where energy and forces are deterministic functions of the coordinates. In coarse-grained systems, the same coordinates generate stochastic energies and forces. The software was implemented using PyTorch Geometric[89] and PyTorch lightning framework[90] and is publicly available in TorchMD-Net[77]. The SchNet architecture has several distinct components, each playing an important function in predicting system forces and energies for given input configurations. The formal inputs into the network are the Cartesian coordinates for a full configuration and a predetermined type for each coarse-grain bead. In the first network operation, a molecular graph, $\mathcal{G}$, is constructed, where each coarse grain bead represents a node. Each node is given an embedding feature vector, the set of which is grouped into a feature tensor. For SchNet, the embedding is produced by applying a learnable linear mapping. The edges of $\mathcal{G}$ are used to define the network operations that update the features of each node. These updates are encompassed in so-called "interaction blocks", which are a form of message-passing updates. The edges of $\mathcal{G}$ are the set of pairwise distances for each bead from its nearest neighbors, the range of which is set uniformly for all beads by an upper cutoff distance. In this way, several interaction blocks can be stacked in

succession to give the network increased expressive power. After the final interaction block, an output network is used to contact the node feature dimension to a scalar for each node. This forms a set of scalar energy predictions, $U$ from each node. By applying a gradient operation with respect to the network input coordinates, the curl-free Cartesian forces, $\mathbf{F}$, are predicted for each bead, representing the final network output.

The hyperparameters were selected based on the quality of the simulation produced using protein-specific models. An example of a training input file is presented in Supplementary Listing 1. The test loss was not a useful metric for hyperparameter selection because the value did not change much between successful and failed models[91,92]. The only way to correctly validate the models was to use them in coarse-grained simulations. A high-quality model produces stable MD simulations, the trajectories explore the conformation landscape and the free energy surface is smooth. In addition, a good model will match the results of all-atom simulations and form energy minima around the relevant states, and we will observe multiple transitions between these states. The hyperparameter combination had the biggest influence on the stability of the MD simulation, the smoothness of the free energy landscape, and the visited areas of the conformational landscape. We found that reducing the number of radial base functions from 150, as in the previous work[63], to around 18 has a big impact on the stability of the MD simulations of proteins bigger than chignolin. With a higher number of RBFs, the forces become spiky for the conformations that are not present in the training set, which leads to the instability of MD runs. Further improvement can be made by replacing the Gaussian function that was used in previous works with expnorm. It is slightly elongated towards longer distances and this shape better suits modeling the properties of CG beads. The smoothness of the landscape was affected the most by the type of activation function. Hyperbolic tangent (tanh) makes the free energy surface smooth, while shifted softplus (ssp) caused the trajectories to collapse into many local minima, making the surface grainy. Other parameters that have a significant influence on the quality of the models are the number of interaction layers and the range of radial base functions. It is important to mention that in some cases even the random seed has an influence on the quality of the models, especially on the coverage of the free energy landscape. For that reason, to ensure reproducibility of the results we trained 2–4 replicas of each model, as mentioned in the main text. Based on the results for protein-specific models as well as the multi-protein model, we selected the following combination: 4 interaction layers, 128 filters used in continuous-filter convolution, 128 features to describe atomic environments, and 18 expnorm as radial base functions (RBF) span in the range from 3.0 to 12.0 Å. In general, we found that the models with hyperparameters similar to these tend to be good quality in terms of the metrics mentioned before.

## Neural network architecture

The series of full network operations can be written as:

$$\boldsymbol{\xi}^0 = \mathbf{W}^E \mathbf{z} \tag{2}$$

$$\boldsymbol{\xi}^1 = \boldsymbol{\xi}^0 + \mathbf{W}^0 \sigma\left(\mathrm{Aggr}\left(\mathbf{W}^C * \boldsymbol{\xi}^0\right)\right) \tag{3}$$

$$\boldsymbol{\xi}^2 = \boldsymbol{\xi}^1 + \mathbf{W}^1 \sigma\left(\mathrm{Aggr}\left(\mathbf{W}^C * \boldsymbol{\xi}^1\right)\right) \tag{4}$$

$$\vdots \tag{5}$$

$$\boldsymbol{\xi}^N = \boldsymbol{\xi}^{N-1} + \mathbf{W}^{N-1} \sigma\left(\mathrm{Aggr}\left(\mathbf{W}^C * \boldsymbol{\xi}^{N-1}\right)\right) \tag{6}$$

$$U = H_{\text{out}}(\boldsymbol{\xi}^N) \tag{7}$$

$$\mathbf{F} = -\operatorname{grad}(U, x) \tag{8}$$

for $N$ interaction blocks. Note that for clarity, we have omitted learnable additive biases in all linear operations above, though they are easily incorporated. The first step of the message-passing update involves expanding the pairwise distances into a set of radial basis functions, $\phi$. $\phi$ then comprises a filter generating network used to produce a set of continuous filters, $\mathbf{W}^C$:

$$\mathbf{W}^C = \mathbf{W}2(\sigma(\mathbf{W}1\phi)) \tag{9}$$

where $\mathbf{W}_1, \mathbf{W}_2$ are learnable linear weights and $\sigma$ is an element-wise non-linearity. These filters are used in a continuous filter convolution through an element-wise multiplication with the current node features for $\mathcal{G}$. These convolved features are then passed through a non-linearity and added directly to the unconvolved node features through a residual connection:

$$\boldsymbol{\xi}^{i+1} = \boldsymbol{\xi}^i + \mathbf{W}^i \sigma\left(\operatorname{Aggr}\left(\mathbf{W}^C * \boldsymbol{\xi}^i\right)\right) \tag{10}$$

where "Aggr" is a chosen pooling/aggregation function that reduces the convolution output (eg, sum, mean, max, etc.). This message-passing update, combined with the residual connection, forms the entirety of an interaction block, producing an updated set of node features for $\mathcal{G}$ that can be used as input for another interaction block. Our implementation of this network architecture allows for training on multiple GPUs and more efficient utilization of GPU memory.

## Coarse-grained simulations

Coarse-grained representations were created by filtering all-atom coordinates such that only certain atoms are retained. This mapping is a simple linear selection, wherein the mapping matrix that transforms the all-atom coordinates to the coarse-grained coordinates is a matrix where zero-entries filter out unwanted beads. The all-atom trajectories were filtered to retain the coordinates and forces of $\alpha$-carbon atoms (CA). To speed up the training of protein-specific modes, trajectories were further reduced by selecting every 10th frame. However, for smaller proteins (Chignolin, Trp-Cage, BBA, and Villin), the training data was not sufficient to produce satisfactory models. Therefore all the frames were used in the training of these systems. To train the multi-protein model we combined the datasets for all the targets. Each CA bead was assigned a bead type based on the amino acid type. In the assignment, we ignored the protonation states and distinguished norleucine, a non-standard residue appearing in Villin, as a unique entity. The terminal residues were assigned the same embedding as the non-terminal residues, despite the charge. As a result, we obtained 21 unique bead types. To each bead type we assigned a unique integer, an embedding that will be used as an input for the network.

To perform the coarse-grained simulations using a trained NNP, we used TorchMD[63], an MD simulation code written entirely in PyTorch[93]. The package allows for an easy simulation with a mix of classical force terms and NNPs. The parameters for the prior energy terms were enumerated and stored in YAML files. The NNP was introduced as an external force, as described in the previous work[63]. We carried out CG simulations over all the proteins, both for each protein-specific model and for the multi-protein model, as well as selected mutants. Simulations were set up with a configuration file (an example in Supplementary Listing 2). We selected 32 conformations evenly distributed across the free energy surface of the reference simulations from where to start the coarse-grained simulations (Supplementary Fig. 1). For each system, 32 parallel, isolated trajectories were run at 350 K for the time necessary to observe transitions

between states with a 1 fs time step, saving the output every 100 fs. The length of each individual trajectory was 1.56 ns (accumulated time of 50 ns) for Chignolin and BBA, 3.12 ns (accumulated time of 100 ns) for Trp-Cage and Villin, 12.5 ns (accumulated time of 400 ns) for WW-Domain and Protein G, and 6.25 ns (accumulated time of 200 ns) for the remaining protein targets. For some systems and models, we were able to obtain stable trajectories with a time step as high as 10 fs. However, to make the results comparable we adapted identical parameters for all simulations, and thus we were limited by the highest possible time step where all types of simulations were stable (1 fs). We observed that for the conformations not represented in the training set, the forces tend to form spikes, which leads to the instability of the simulations. This can be reduced by applying prior force terms and applying cutoffs to radial base functions that limit the exploration of unphysical conformations. In addition, a reduced number of radial base functions has a positive impact on the overall smoothness of the force field. The coarse-grained simulations were performed using Nvidia GeForce RTX 2080 graphics cards.

## Prior energy terms

The pairwise bonded term was represented with the following equation:

$$V_{\text{bonded}}(r) = k(r - r_0)^2 + V_0, \tag{11}$$

where $r$ is the distance between the beads forming the bond, $r_0$ is the equilibrium distance, $k$ is the spring constant and $V_0$ is a base potential. The nonbonded repulsive term was represented by the potential

$$V_{\text{repulsive}}(r) = 4\epsilon r^{-6} + V_0, \tag{12}$$

where $\epsilon$ is a constant that was fit to the data, $r$ is the distance between the beads and $V_0$ is a base potential. The parameters were used as in TorchMD[63]. The parameters for norleucine, a non-standard residue appearing in Villin, were adapted from leucine. In addition, we introduced a third prior dihedral term:

$$V_{\text{dihedral}}(\phi) = \sum_{n=1,2} k_n(1 + \cos(n\phi - \gamma_n)), \tag{13}$$

where $\phi$ is the dihedral angle between the four consecutive beads, $k_n$ is the amplitude and $\gamma_n$ is the phase offset of the harmonic component of periodicity $n$. The parameters for dihedral terms were fit to the data used for training, containing all the proteins. The extracted values of $k_n$ were scaled by half to achieve a soft prior that will break the symmetry in the system but will not disturb the simulation in a major way. For simplicity, all combinations of four beads were treated equally, therefore all dihedral angles were characterized by the same set of parameters, in contrast to bonded and repulsive prior. The force field file with terms and associated parameters is available in the GitHub repository. To enable the simultaneous use of both Dihedral and RepulsionCG force terms in TorchMD, exclusions between pairs of beads for RepulsionCG term are defined by an additional parameter "exclusions".

## Markov state model estimation and structure selection

For the analysis of the CG simulations and their comparison with the all-atom MD simulations, we built MSMs for each protein, both for the all-atom MD simulations and the two sets of coarse-grained simulations (protein-specific and multi-protein models). The basic concept behind MSMs is that the dynamics of the system are modeled as a memory-less jump process, where future states are only conditioned on the current state, hence the dynamics are Markovian. MSM estimation of transition rates and probabilities requires partitioning the high-dimensional conformational space into discrete

states. In order to project the high-dimensional conformational space into an optimal low-dimensional space, we use TICA, a linear transformation method that projects simulation data into its slowest components by maximizing autocorrelation of transformed coordinates at a given lag time[70,71]. The resulting low-dimensional projected space is then discretized using a clustering algorithm for the MSM construction.

For the all-atom MD simulations, we featurized the simulation data into pairwise $C_\alpha$ distances and applied TICA to project the featurized data into the first 4 components. Next, the components were clustered using a K-means algorithm and the discretized data was used to perform the MSM estimation. Although better reference MSM models could be obtained by using different featurizations, we are limited to only using pairwise $C_\alpha$ distances as it is transferable between systems and comparable with the coarse-grained simulations.

For the coarse-grained simulations, the same procedure was used. However, when projecting the featurized data into the main TICs, we used the covariance matrices computed with the all-atom MD simulations to project the first 3 components, in order to compare how well the coarse-grained simulations reproduce the free energy surface for each protein. For each MSM, we used the PCCA algorithm to cluster microstates into macrostates for better interpretability of the model and to define a native macrostate that we can use to evaluate the performance of the coarse-grained simulations.

The free energy surface plots used for comparison were obtained by binning over the first two TICA components, dividing them into an 80 × 80 grid, and averaging the weights of the equilibrium probability in each bin, obtained for each defined microstate through MSM analysis. To recover the native conformation from a set of coarse-graining simulations, we used the MSMs and sampled conformations from the native macrostate. The native macrostate was defined as the macrostate containing the frame with the minimum RMSD to the experimental structure.

### Statistics and reproducibility

To ensure the reproducibility of the results, the training of each model was repeated 2 to 4 times with different random seeds. Each replica was then tested by performing a fast simulation of 4 parallel trajectories of a corresponding system, with the objective of a fast assessment of the model. The model that produced the best results was selected for the main validation.

All the statistics obtained using MSMs are reported with an average and standard deviation obtained from estimating 10 different models by bootstrapping the simulation data, taking 90% of the trajectories. This was performed for both reference MD and coarse-grained simulations. In addition, for coarse-grained trajectories, we removed 10% of the initial frames of each trajectory from the analysis to avoid biasing the model with starting conformations. All structures shown were obtained by sampling 10 conformations from the corresponding macrostates.

### Reporting summary

Further information on research design is available in the Nature Portfolio Reporting Summary linked to this article.

## Data availability

All relevant data supporting the key findings of this study are available within the article and its Supplementary Information files. The models and data generated in this study are available at github.com/torchmd/torchmd-protein-thermodynamics (https://doi.org/10.5281/zenodo.8155342)[94]. Starting structures for molecular dynamics were sourced from Protein Data Bank https://www.rcsb.org/, with PDBids: 5AWL, 2JOF, 1FME, 2F4K, 1PIN, 2HBA, 2WXC, 1PRB, 1ENH, 1MI0, 2A3D, 1LMB, 1BAL, 1GAB, 2N35, 1DU0, 1P7I, 1P7J, 2HOS, 6M3D, 2MTQ, 1LLI, 3KZ3. Source data are provided with this paper.

## Code availability

All codes are free and available in github.com/torchmd. The code to run molecular dynamics is available at github.com/torchmd/torchmd(https://doi.org/10.5281/zenodo.8155115)[95]. The neural network architecture is available at github.com/torchmd/torchmd-net(https://doi.org/10.5281/zenodo.8155330)[96].

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

## Acknowledgements

The project PID2020-116564GB-I00 has been funded by MCIN/AEI/10.13039/501100011033 (G.D.F.) This project has received funding from the Torres-Quevedo Program from the Spanish National Agency for Research (PTQ2020-011145/AEI/10.13039/501100011033) (M.M.); the Torres-Quevedo Program from the Spanish National Agency for Research (PTQ2021-011669/AEI/10.13039/501100011033) (A.P.); the European Union's Horizon 2020 research and innovation program under grant agreement No. 823712 (G.D.F.); NLM Training Program in Biomedical Informatics and Data Science (grant no. 5T15LM007093-27) (N.E.C.); Deutsche Forschungsgemeinschaft (DFG, GRK DAEDA-LUS, RTG 2433, Project Q05) (N.E.C.); National Science Foundation (CHE-1900374 and PHY-2019745) (C.C.); Einstein Foundation Berlin (Project 0420815101) (C.C.); Deutsche Forschungsgemeinschaft (DFG) SFB 1114 projects A04, B03, and B08, SFB/TRR 186 project A12, and SFB 1078 project C7 (C.C.); Deutsche Forschungsgemeinschaft (DFG) projects CRC1114/A04, CRC1114/C03 (F.N.); European Research Council (ERC) project ERG CoG 772230 (F.N.); Berlin Mathematics Center MATH+ project AA1-6 (F.N.); the National Institute of General Medical Sciences (NIGMS) of the National Institutes of Health under award number GM140090 (G.D.F.). The content is solely the responsibility of the authors and does not necessarily represent the official views of the National Institutes of Health. The authors thank the volunteers of GPUGRID.net for donating computing time. T.G. acknowledges the CINECA award under the ISCRA initiative, for the availability of high performance computing resources and support. This project has received funding from the Spoke 7 of the National Centre for HPC, Big Data and Quantum Computing (CN00000013) of the NextGenerationEU initiative (T.G.).

## Author contributions

M.M. and A.P. contributed equally to this work. G.D.F., F.N. and C.C. conceived the presented idea. G.D.F., M.M., and A.P. designed the study. M.M. and A.P. carried out the simulations and analyzed the data. N.E.C., T.G. and B.E.H. helped to analyze and interpret the data. S.D., P.T. contributed software used in the work. M.M., A.P. and G.D.F. wrote the manuscript. All authors provided critical feedback and helped shape the research, analysis and manuscript.

## Competing interests

The authors declare no competing interests.
