## [Peer Review File · Nature Communications]

Machine Learning Coarse-Grained Potentials of Protein ThermodynamicsEditorial Note: This manuscript has been previously reviewed at another journal that is not operating a transparent peer review scheme. This document only contains reviewer comments and rebuttal letters for versions considered at *Nature Communications*.

Reviewer #1 (Remarks to the Author):

Minor revision required.

My concerns regarding the results of this manuscript have been mostly addressed. However, the authors should explicitly refer to and discuss reference 58 of the current manuscript as a reasonably successful previous investigation towards transferable bottom up CG models of proteins. It is not proper scholarship to avoid giving due credit to this earlier research nor to ignore my prior reviewer concerns over this omission.

Reviewer #2 (Remarks to the Author):

The paper by Majewski, Perez et al is a revised version of a paper I reviewed for a different journal. For the sake of readers that might be able to access the review for this journal, I provide a slightly revised version of some of my comments on the previous paper further below. Before doing so, I will repeat that I think that this paper is excellent. It provides an honest description of what we can and cannot yet do with machine-learned neural-network potentials. The results are interesting, and while they are rather impressive from a technical point of view, they also show that we are not yet at a stage where these can provide much additional information beyond that available in the training data. I stress, however, that we need studies of this kind and high quality to know where the field is.

I appreciate the authors' honest description and answers, and think that the changes to the paper have made it even clearer what has been achieved and what we cannot yet do. My only remaining suggestion is to change "thatt" in line 263 to "that".

Below is a slightly revised version of general comments I made on a previous version of the paper: The paper by Majewski, Perez et al presents an interesting attempt to generate a coarse-grained potential for molecular dynamics (MD) simulations of proteins. Briefly described, the work involves generating a dataset of all-atom MD simulations of protein folding and training a neural-network potential (NNP) for a coarse-grained (CG) model using the all-atom data as target data. Briefly summarized, the results show that even when using state of the art methods and training data, we do not appear to be close to having models that generalize to unseen proteins or across the entire free energy landscape.

The work stands out by being very well done and by attacking an important problem. Also, the authors are commended for presenting their results in a sober tone. The results, while interesting and forward looking, are perhaps not as impressive as one might have hoped. That is, while the authors used state-of-the-art training data and state-of-the-art methods for constructing and training NNPs, the resulting models do not generally capture all of the training data and appear not to generalize well to proteins outside the training set. If one takes such generality as the goal, then the results are perhaps "negative". On the other hand, it is nice to see that one can train a model that at least captures some/many aspects of the training data.

One thing I am missing from the work is some forward-looking assessment of what we are missing. Obviously, there is no guarantee that it is possible to construct a CG potential that can fold proteins or match the free-energy landscapes from all-atom MD. But from the work, it is still not clear whether the limitation is from the model or the training.

The authors conclude by suggesting that with a much larger and structurally diverse set of MD data, it might be possible to train a general (transferable) NNP. I, however, do not see clear evidence of this, and it would have been useful with a deeper analysis of this point. How far are we from a potential that transfer to proteins outside the training data? Would we need 20 proteins, 100, or 1000? And how would one generate the training data given the scarcity of a diverse set of fast folding proteins that can be folded by current all-atom potentials and sampling methods?

Related to the above, the approach taken (at least at the moment) requires the authors to study

fast-folding proteins. Since these, generally, have smoother landscapes and fewer kinetic traps/intermediates than "normal" proteins, it is not clear that the training procedure has the chance to capture such non-native states. While there are hints at intermediates captured in the trained models (for beta sheet proteins), this does not appear to be generally true. This then leads to the question of how the procedure outlined may be extended to ensure that the model captures more than "just" the native state. I write "just" because obviously having a CG potential that could fold proteins would be a tremendous step forward, but on the other hand predicting the overall native state is now mostly a solved problem. So, as the authors state, the outstanding problem is to get the free energy landscapes/thermodynamics. Thus, I would have liked to see a clearer analysis of how well the model captures intermediates. In which ways would this kind of model be better than taking an AlphaFold structure and using it to construct a Go-model? I am not saying that the model could not be useful, but rather that I am lacking some clearer and feasible goals of the approach.

Reviewer #3 (Remarks to the Author):

The authors have done significant efforts to revise the manuscript. In particular, I am happy with some of the claims being toned down relating to model transferability. I believe the current state of the manuscript is suitable for publication.

Reviewer #1
Reviewer #2
Reviewer #3

2
3
4

Reviewer #1

We thank the reviewer for the useful feedback. To address the latest comment, we mentioned the work of Hills *et al.* in the introduction along with other important works from the field of coarse-graining.

“Notably, the work by Hills *et al.* has made significant strides towards creating a transferable bottom-up coarse-grained potential for the simulation of proteins, contributing valuable insights to the field.”

All the changes are marked in red.

Reviewer #2

We thank the reviewer for the valuable feedback. The comments helped us to improve the manuscript significantly.

To address the last suggestion we corrected the typo in line 263 from “thatt” to “that”. The correction has been highlighted in red.

Reviewer #3

We thank the reviewer for their feedback.